# Comparison of Long-Read Methods for Sequencing and Assembly of Lepidopteran Pest Genomes

**DOI:** 10.3390/ijms24010649

**Published:** 2022-12-30

**Authors:** Tong Zhang, Weiqing Xing, Aoming Wang, Na Zhang, Ling Jia, Sanyuan Ma, Qingyou Xia

**Affiliations:** 1State Key Laboratory of Silkworm Genome Biology, Biological Science Research Center, Southwest University, Chongqing 400715, China; 2Chongqing Key Laboratory of Sericulture Science, Chongqing Engineering and Technology Research Center for Novel Silk Materials, Chongqing 400715, China

**Keywords:** biological control, de novo assembly, long-read sequencing, benchmarking, lepidopteran pest, assembly evaluation

## Abstract

Lepidopteran species are mostly pests, causing serious annual economic losses. High-quality genome sequencing and assembly uncover the genetic foundation of pest occurrence and provide guidance for pest control measures. Long-read sequencing technology and assembly algorithm advances have improved the ability to timeously produce high-quality genomes. Lepidoptera includes a wide variety of insects with high genetic diversity and heterozygosity. Therefore, the selection of an appropriate sequencing and assembly strategy to obtain high-quality genomic information is urgently needed. This research used silkworm as a model to test genome sequencing and assembly through high-coverage datasets by de novo assemblies. We report the first nearly complete telomere-to-telomere reference genome of silkworm *Bombyx mori* (P50T strain) produced by Pacific Biosciences (PacBio) HiFi sequencing, and highly contiguous and complete genome assemblies of two other silkworm strains by Oxford Nanopore Technologies (ONT) or PacBio continuous long-reads (CLR) that were unrepresented in the public database. Assembly quality was evaluated by use of BUSCO, Inspector, and EagleC. It is necessary to choose an appropriate assembler for draft genome construction, especially for low-depth datasets. For PacBio CLR and ONT sequencing, NextDenovo is superior. For PacBio HiFi sequencing, hifiasm is better. Quality assessment is essential for genome assembly and can provide better and more accurate results. For chromosome-level high-quality genome construction, we recommend using 3D-DNA with EagleC evaluation. Our study references how to obtain and evaluate high-quality genome assemblies, and is a resource for biological control, comparative genomics, and evolutionary studies of Lepidopteran pests and related species.

## 1. Introduction

Lepidopteran pests have a critical impact on vegetable crop production, often with a mixture of multiple pests, with many overlapping generations each year, causing huge annual economic losses. Genome sequencing has brought Lepidopteran pest control and genomics research to a new level. Genome sequencing of *Plutella xylostella*, *Hyphantria cunea*, *Cydia pomonella*, and *Helicoverpa zea* revealed the genetics of their invasive populations, explained their host and environmental adaptations at the genetic level, provided partial evidence for the causes of their rapid invasion, and determined potential genetic targets for innovative pest management strategies and the genetic basis of Bt toxin resistance [1,2,3,4,5]. Whole-genome sequencing of twenty *Heliconius* butterflies revealed the complex evolutionary history of the genus, demonstrating that chromosomal structural variation due to gradual penetration is responsible for increased polymorphism in butterfly wings [6].

Since the publication of the first Lepidopteran pest genome [7], sequencing and analytical technologies have developed rapidly. The emergence of innovative subversion techniques, such as short-read sequencing, long-read sequencing, link readout, High-through chromosome conformation capture (Hi-C), optical mapping, and different assembly methods hugely promote genome assembly [8,9]. It is expensive to carry out genome assembly in non-model organisms, and the draft is usually constructed by thousands of fragmented contigs and scaffolds. Nowadays, for many genome projects, achieving high continuous and quality assembly that is close to the chromosome level is a realistic and affordable goal. More than 204 Lepidopteran pests have been sequenced at the nuclear genome level and made publicly available [10]. However, there have been disparate genome sequencing efforts in Lepidopteran pests and many orders remain without genomic representation [11]. With the advent of the pan-genomic era, more Lepidoptera will be sequenced in the future.

The continuity of de novo genome assembly was greatly improved by long-read DNA sequencing platforms, such as single molecular real-time (SMRT) sequencing, Oxford Nanopore Technologies (ONT), and Pacific Biosciences (PacBio) [12]. These technologies overcome the shortcomings of next-generation DNA sequencing (NGS), including information loss, sequence-dependent biases, and relatively short-reads [13]. Previous studies compared the genome assembly tools of ONT sequencing datasets or HiFi sequencing datasets used in Escherichia coli, viruses, pathogens, yeast, fruit flies, and rice, and predominantly used simulated datasets to construct low-quality assembly [14]. In the research of Lepidopteran pests, there has been no large-scale analysis and evaluation of genome assemblers based on high-depth third-generation sequencing (TGS) datasets, and no complete melanosome-to-melanosome genome assembly, which greatly limits the functional research and pest control of lepidopteran insects. Therefore, there remains an urgent need to choose an appropriate sequencing platform, advanced genome assembler, and sequencing depth for the investigation of Lepidopteran pests.

However, genome quality assessment is also very important. Assembly errors are not always apparent and can inadvertently lead to fictitious conclusions [15,16]. The contig and scaffold N50 were used for measuring the fragmentation degree of genome assembly, and Benchmarking Universal Single-Copy Orthologues (BUSCO) is currently used for evaluating the representation of genes [17]. Recently, new methods for assessing the quality of genome assemblies have emerged, such as QUAST-LG, Merqury, KAT, and Inspector [18]. Hi-C technology was used to study three-dimensional (3D) genomic architectures and now has been used for draft genome assembly improvement and chromosome scaffolding in large genomes [19]. Meanwhile, the quality of genome assemblies can be assessed using Hi-C interaction heat maps, the assembly errors usually appear in the chromatin interaction breakpoints. When mapping the Hi-C data to the reference genome, aberrant interaction blocks with different orientations represented different types of assembly errors. However, these methods are achieved by aligning the sequencing reads to contigs. Although Inspector has made improvements in its algorithm to reduce the runtime, it is generally not a particularly rapid method, and there is short of effective tools to accurately evaluate chromosome-level genome assembly, large structural errors in particular.

*Bombyx mori* (*B. mori*) is a good model for genome assemblies evaluation, as many genome datasets are readily available to benchmark the completeness and accuracy of assemblies [20]. Considering the high genome heterozygosity of field-collected Lepidopteran pests limited by time and space, the genome is at risk of degradation if it cannot be extracted in a timely manner. In this study, we performed 32 (four assemblers on eight subsets with different sequencing depths), 42 (six assemblers on seven subsets with different sequencing depths), and 12 (two assemblers on six subsets with different sequencing depths) de novo assemblies on high-coverage ONT, PacBio continuous long-reads (CLR) and HiFi datasets, respectively. These were performed for three silkworm strains D9L × N4, D9L, and P50T, corresponding to three conditions: highly heterozygous, degradation, and normal. The quality of assembly was evaluated by QUAST [21], BUSCO, Inspector, and the newly proposed EagleC [22] based on deep learning. We assessed the performance of diverse TGS approaches in *B. mori*, focusing on how to efficiently and accurately construct and evaluate chromosome-level genome assemblies in *B. mori* and other Lepidopteran pests. We believe our results will provide valuable guidance for future Lepidopteran pest genome projects as well as improve previous genome assemblies without generating new sequencing data.

## 2. Results

### 2.1. Summary of Raw Data, Assemblies, and Benchmarks

To compare the performance of diverse TGS platforms on constructing highly contiguous genome assembly on Lepidopteran pests. We sequenced and analyzed three long-read datasets for three *B. mori* strains (Table 1 and Figure 1): (1) Silkworm D9L, PacBio CLR reads, 48 Gb data (110×, N50 = 11,722 bp, E-size = 11,909 bp), (2) Silkworm P50T, PacBio HiFi reads, 27 Gb data (60×, N50 = 15,818 bp, E-size = 16,484 bp), and (3) Silkworm D9L × N4, ONT reads, 70 Gb data (160×, N50 = 32,103 bp, E-size = 33,543 bp).

The genome sequencing datasets were assembled by seven different assembly tools (Figure 2). The CLR reads were assembled by Canu, NextDenovo, MECAT, and wtdbg2. Assemblies of HiFi reads were performed using HiCanu and hifiasm. ONT data were assembled using NextDenovo, NECAT, and wtdbg2.

The assembly quality was evaluated according to the following six criteria: contig numbers (Contigs), contig N50 (N50) length, number of structural errors (Structural error), small structural errors per Mb (Small-scale error), number of BUSCO complete genes (Complete genes), Quality Value (QV) score and percentage of assembly errors (PAR) identified by EagleC for chromosome-level genomes.

### 2.2. ONT Genome Assembly

For investigating how to obtain high-quality haploid genome assemblies for field-caught genomically heterozygous Lepidopteran pests, we selected the silkworm D9L × N4 strain (approximately 1.11%, Appendix A) with high genomic heterozygosity for ONT sequencing and assembly testing. The ONT sequence was assembled using three different long-read assembly tools (NextDenovo, wtdbg2, and NECAT) and eight different subsets of various coverage (10×, 20×, 40×, 60×, 80×, 100×, 120× and 160×). Detailed statistics for each assembly are shown in Table 2 and Appendix A.

The NextDenovo assemblies were the smallest in size (approximately 449–468 Mb) with contig numbers of approximately 89–114 (Figure 3 and Appendix A). NextDenovo generated the most contiguous assemblies (contig N50 approximately 10.0–13.8 Mb), with the highest number of complete (approximately 1181–1298) and single-copy (approximately 1176–1287) BUSCO genes. The wtdbg2 assemblies were the largest in size (approximately 452–794 Mb) and produced contig numbers of approximately 3273–13,714. Wtdbg2 generated the least contiguous assemblies (contig N50 0.15–0.81 Mb), and the lowest number of complete (669–1129) and single-copy (668–1083) BUSCO genes. The assembly quality of wtdbg2 for genomes with high heterozygosity was less satisfactory, but it is the only software that can generate assembly at 10× sequencing depth. The assembly quality of NECAT was between those of NextDenovo and wtdbg2. The NECAT assemblies’ sizes were approximately 561–581 Mb, contig numbers were approximately 688–851, contig N50 were between 2.44 and 2.88 Mb, and complete BUSCO genes were between 1253 and 1272. Additionally, we analyzed the computational time of those assemblers and found that the wtdbg2 was the fastest assembler, followed by NextDenovo and NECAT (Figure 4b), saving between a third and a half of the time when the sequencing depth was greater than 80×.

To estimate the genome assembly accuracy, we calculated the number of Structural errors and Small-scale errors using Inspector. NextDenovo had the lowest number of Small-scale errors, and Structural error numbers just below those of wtdbg2 (Figure 5 and Appendix A). Wtdbg2 had the highest number of Small-scale errors, while the lowest number of Structural errors. NECAT had the highest number of Structural errors and the second highest of Small-scale errors.

The subsequent Racon long-read polishing process greatly improved the wtdbg2 draft genome assemblies’ completeness as indicated by the BUSCO complete gene percentages, which increased from between a minimum range of 49% to 82.6% to a maximum range of 74.7% to 92.1% (Appendix A). The assembly accuracy was also greatly improved as indicated by the number of Small-scale errors, which decreased from between 3484 and 7767 per Mbp to between 633 and 1418 per Mbp (Appendix A).

For the purpose of investigating the effect of sequencing depth on assembly tools, we evaluated the quality of ONT assemblies on diverse sequencing depths (10×, 20×, 40×, 60×, 80×, 100×, 120× and 160×). The assembly quality on low-depth subsets (10× or 20×) varied greatly amongst different assemblers, whereas it was reliable on relatively high-depth subsets (Figure 3). According to our findings, the dataset with around 40× ONT can construct the most genomes. However, a deeper sequencing effort is required to further enhance the genome quality.

### 2.3. CLR Genome Assembly

In order to investigate how to generate high-quality haploid genome assemblies from wild harvested genome degradation samples, we selected the slightly poorer quality genomes extracted from silkworm D9L samples for testing (N50 = 11,722 bp, E-size = 11,909 bp, Table 1 and Figure 1). The assembly of the CLR reads was conducted using four different long-read assembly tools (NextDenovo, Canu, wtdbg2, and MECAT2). Due to lower genomic heterozygosity, the CLR assemblies showed much smaller differences than ONT in genome size (426–506 Mb, excluding the 10× and 20× results, Appendix A). When a certain sequencing depth is satisfied (>= 40×), the difference in the number of contigs for each genome assembly is not significant, and the result of NextDenovo remains the best. The continuity of all the assemblies (N50 of contigs) increased by the sequencing depth, the NextDenovo assembly increased the most pronounced (Figure 3).

The CLR assemblies showed similar complete BUSCO gene numbers as the ONT assemblies (excluding MECAT2). Wtdbg2 generated the lowest number of Structural errors, followed by NextDenovo (Figure 5). NextDenovo generated the lowest number of Small-scale errors followed by Canu (Appendix A). The NextDenovo assembly showed the highest contiguous (contig N50 = 9.41 Mb), smallest size (477 Mb), and the least contigs (n = 205) (Appendix A). The Canu assembly was the largest (506 Mb) but contained a high degree of duplication as indicated by the percentage of duplicated BUSCOs (2.9%). Therefore, as was recently discovered, the assembly of Canu probably contains uncollapsed haplotypes corresponding to artifactually duplicated areas [23]. The assembly quality of four assemblers was assessed by the metric mean of the six different subsets (Table 2). NextDenovo shows the best overall performance, followed by Canu. Though Canu needs the longest CPU hours and generates fragmented assemblies, the accuracy is excellent (Figure 4b).

Before polishing, the wtdbg2 assembly was the most fragmented (contig N50: 0.154–2.56 Mb) and the MECAT2 assembly was the least complete (12.5–57.8% complete BUSCOs) (Appendix A). Subsequently, we polished the wtdbg2 and MECAT2 assemblies using the CLR long-reads. The Racon polishing steps greatly improved the wtdbg2 and MECAT2 draft assemblies’ genome completeness (Figure 3). As expected, a reduced number of Small-scale errors and Structural errors were identified in CLR assemblies when compared with ONT assemblies (Figure 5 and Appendix A). The long-read polishing process resulted in the percentage of single-copy BUSCOs increasing significantly and the number of Small-scale errors for the MECAT2 assembly reduced sharply, while the number of Small-scale errors of wtdbg2 assembly reduced modestly (Appendix A). Interestingly, the long-read polishing process did not improve the integrity of the NextDenovo and Canu assemblies. In this part, we found that the dataset with roughly 40× CLR can construct most genomes, increase the sequencing depth could improve the genome quality, and need a polishing strategy or not depending on which assemblers are used.

### 2.4. HiFi Genome Assembly

Purposed to testing the contribution of the new technology to high-quality genome assembly, we performed PacBio HiFi sequencing on P50T silkworm whose genomes had been previously sequenced. HiFi reads were assembled using HiCanu and hifiasm. Compared with CLR and ONT assemblies, the genomic continuity and integrity of HiFi assemblies were significantly superior. There were no significant differences in the size, continuity, and completeness of the HiFi genome assemblies. The greatest difference is reflected in the contig numbers, which are much smaller in hifiasm assemblies than those in HiCanu assemblies (Figure 3 and Appendix A). We polished the HiFi assemblies using the HiFi long-read sequences. Just as expected, the polished HiFi assembly showed a similar percentage of complete BUSCO genes compared to the raw HiFi assembly (Appendix A). When compared with the ONT and CLR assembly, the HiFi assembly contained the fewest Structural errors and Small-scale errors (Figure 5, Appendix A).

Furthermore, we evaluated the quality of HiFi assemblies on datasets with different sequencing depths (10×, 20×, 30×, 40×, 50×, and 60×) to investigate the effect of data depth on assemblers. For low-depth subsets (as the 10× subset depicted in Figure 3), the assembly quality was highly varied amongst assemblers, while on relatively high-depth subsets, it was resilient. When exceeding a certain threshold, high coverage subsets do not significantly improve the quality of assembly, either. However, higher depths will require more computing resources and instantiation times. Therefore, choosing an appropriate depth is crucial. According to our findings, the dataset with about 20× HiFi data was able to create the most genomes. Since HiFi only requires a sequencing depth of 20× or more to build most of the genome and does not require a subsequent polish process, the time used for genome assembly is much less than that of ONT and CLR, especially when using Canu.

Compared with the other two sequencing methods, HiFi assembly shows the best assembly quality, the lowest contig number, and the highest continuity, accuracy, and completion, without relying on other scaffolding. It also requires the least amount of time and computer memory and can be considered the optimal sequencing method for future Lepidopteran pest genomes.

### 2.5. Construction and Quality Assessment of Hi-C-Based Chromosome-Level Genomes

The quality of the three long-read sequencing assemblies was significantly superior compared with short-read sequencing. However, none of the HiFi assemblies completed the assembly of all the chromosomes. We selected the best genome assembly for each sequencing method using 3D-DNA for genome construction at the chromosome level. Using default parameters, 3D-DNA achieved clustering of most of the chromosomes. However, there remained some chromosome clustering errors and contig translocations and inversions, which were identified using the Hi-C map (Appendix A). Further manual adjustment by Juicebox can fix these organizational errors. However, this individualized manual adjustment often does not conform to a uniform standard. We then designed a quality assessment standard for chromosome-level genome assembly based on EagleC. This can identify organizational errors rapidly and accurately and is able to report the percentage of misassemblies in the genome assembly in the form of a table to facilitate the correction of these assembly errors (Figure 6c and Table 3). Based on EagleC’s recommendations, we completed the adjustment of the genome assemblies and performed polishing using Racon and gap filling using TGS-GapCloser. Finally, using five-base telomere repeats (‘*TTAGG*’) [24] as a sequence query, we identified 50 telomeres and constructed 28 pseudomolecules (25 of 28 were represented by a single large contig, and the remaining three were assembled from two main contigs) for the silkworm (P50T-HiFi) genome (Figure 6a,c). Compared with the SilkBase reference genome (P50T-SilkBase), the P50T-HiFi assembly filled 30 gaps that were found in the SilkBase assembly. These gaps ranged from 99 to 75,391 bp and were distributed throughout the genome. The parallel plots showed that the P50T-HiFi assembly displayed good collinearity with the P50T-SilkBase assembly in most of the chromosomes, however, we also found some differences (Figure 6c). According to the EagleC report, these discrepant regions are caused by several Mb-level assembly errors, for example Chr24 (Figure 6e). The assembly mistake in the P50T-SilkBase assembly is also confirmed by the Chr19 parallel plots of five silkworm genome assemblies (Figure 6d). Furthermore, the final remaining three gaps in the P50T-HiFi assembly were filled with P50T-SilkBase assembly, resulting in a gap-free silkworm genome assembly (P50T). This is the first nearly complete telomere-to-telomere reference genome of silkworm (P50T). Although the genome assembly quality of CLR and ONT is not as good as that of HiFi, both completed very high consecutive and complete chromosome-level genome assemblies after treatment with EagleC and 3D-DNA, which is based on Hi-C (Figure 6b and Table 3).

### 2.6. Case

Whether our assembly process is applicable to the genome assembly of other Lepidopteran pests, and whether it can help optimize the genome assembly of published genomes, we selected the genome sequencing data of Korean silkworm (KRSM) [25] and *Dendrolimus punctatus* (*D. punctatus*) [26] for testing. Based on the results of the above comparison, an optimal pipeline was selected for those Lepidopteran insects.

We evaluated the metrics of the final assemblies and demonstrated these in Table 3 and Appendix A. The pipeline can build the high-completeness genome assembly in KRSM with approximately 16.89 Mb scaffold N50, 97.5% complete BUSCO genes (Figure 6b), 55.13 small-scale errors per Mb, 34 structural errors and the value of PAR is 0.05%, demonstrating the accuracy of this genome assembly. On the other hand, misassemblies in the genome assembly of *D. punctatus* were identified and optimized using the EagleC evaluation process, significantly improving the quality of the optimized genome (Figure 7b–d). The circle plots showed that the genome of *D. punctatus* we assembled here shared good collinearity with *Dendrolimus kikuchii (D. kikuchii)*, and confirmed the assembly errors published in previous studies (Figure 7e). This demonstrates the compelling potential of the EagleC evaluation process to assess and optimize the quality of published genome assemblies.

For genome sequencing of Lepidopteran pests, we recommend HiFi and Hi-C sequencing followed by hifiasm and 3D-DNA for assembly and chromosome mounting, which achieves the best haploid genome assembly. For species already sequenced by ONT or CLR, we recommend NextDenovo, 3D-DNA, and EagleC for chromosome-level genome optimization.

## 3. Discussion

Lepidoptera is the second largest order of insects, some of which are severe pests of agriculture and forests and cause significant economic losses each year. Genome sequencing of Lepidopteran pests has contributed greatly to their control. The accumulating genomic resources have been a crucial source for major breakthroughs in life science innovations and discoveries. Seventy-six arthropod genome assemblies were used to characterize the changes in genes and protein contents for a better understanding of 500 million years of evolution [27]. A study of 195 insect genomes revealed a high diversity of transposable elements across insects with varying degrees of conservation depending on phylogenetic position [28]. Horizontal gene transfer (HGT) events in 218 insects acquired from non-metazoan sources provide insight into the adaptation of GTs in insects [29]. The breadth of Lepidopteran pest genome sequencing spans approximately 300 million years of evolution and roughly two orders, with genome sizes ranging from the tiny 229.9 Mb genome of *Papilio polytes* to the massive genome of *Parnassius apollo* at 1392 Mb [10]. However, this represents a mere 0.08% of the approximately 160,000 described insects [11], with many orders remaining without genomic representation. In the future, more Lepidopteran pest genomes will be sequenced. There are a number of questions that need to be addressed at that time: What sequencing strategy and which assembler should be chosen? How can we complete the genome assembly and quality assessment faster and more accurately?

In this study, we use the silkworm as a model to compare and analyze the mainstream TGS strategies, including sequencing platform, sequencing depth, and assemblers. Using the silkworm high-quality genome assembly as a reference, we compared BUSCO, Inspector, and EagleC to find the most appropriate long-read sequencing strategy under different conditions. We performed 32, 42, and 12 de novo genome assemblies of silkworms on high-depth ONT, CLR, and HiFi reads, correspondingly to comprehensively assess the effect of assemblers and sequencing coverage on Lepidopteran insect genome assembly. Each sequencing strategy has its own advantages: the ONT genomic library has the largest fragment size, the HiFi data has the highest accuracy, and the CLR is in between. The focus of different assemblers is different.

On graft genome assembly, the NextDenovo assembler shows the best performance on both CLR and ONT sequencing datasets, and hifiasm is better for HiFi datasets. We recommend the use of 3D-DNA in combination with the EagleC evaluation to complete the construction of chromosome-level genomes. The polishing process is completed without the original assembly, you can finish the scaffolding and polish it afterward, the quality of the assembly is similar, and it is timesaving. In the case application, the KRSM genome was obtained from CLR and Hi-C data with excellent continuity and integrity. Other factors, such as throughput, convenience, and price should also be reasons for considering which genome sequencing platform to choose.

Here, using various techniques and ultra-high-depth datasets, we evaluated the effects of sequencing coverage on Lepidopteran pest genomes assembly. The quality of the genome tends to stabilize after 40× on the ONT and CLR datasets, and at 20× on HiFi datasets, and improves as the sequencing depth increases. The sequencing depth of 20× is the minimum for genome construction, however, the higher the sequencing depth, the better the assembly effect is not necessary, and too high a sequencing depth will cause excessive consumption of computing resources. Especially for large genome projects, such as pan-genome, you must select an appropriate sequencing depth to efficiently reduce the burden of computing resources and the cost of time and money.

The currently available mainstream genome quality assessment markers are N50, BUSCO, Merqury, QUAST-LG, and Inspector. However, the N50 is just a simple continuous statistic, BUSCO can only evaluate conserved genomic regions, Mercury requires users to input high-precision reads and is not suitable for long-read data, and QUAST-LG relies excessively on existing reference genomes [18]. Several software programs have been developed to implement scaffolding based on Hi-C data, HiRISE, LACHESIS, SALSA, 3D-DNA, and ALLHiC [30]. EagleC combines deep-learning and ensemble-learning strategies can predict the whole range of SVs with a Hi-C map very quickly and accurately [22]. SVs can induce de novo chromatin interactions across the breakpoints, which are similar to assembly errors, both show aberrant interaction blocks. We then designed a quality assessment standard for chromosome-level genome assembly based on EagleC. The EagleC process that we developed is different from the previous evaluation work. It is a deep learning-based process used to accurately and rapidly evaluate the quality of chromosome-level genome assemblies and direct the repair of assembly errors. It can also be used for the optimization of published chromosome-level genome assemblies. The quality of de novo genome assemblies has a great significant impact on gene annotation and comparative genomic research [31]. At the same time, we noticed that although we have developed good quality assessment standards for chromosome-level genome assembly, it is difficult for novices to complete the assessment. Building an online database that can generate results reports with one click can greatly solve this problem, and facilitate the widespread use of scholars with different research backgrounds, which is what we are currently doing.

In this study, we take into account that practical issues are faced in Lepidopteran pest genome sequencing projects, including high genome heterozygosity, poor quality genome libraries with short fragments, and assembly approaches tailored to various scenarios. Additionally, it has demonstrated how to improve an existing genome assembly based on the findings of a genome evaluation without producing new sequencing data. This benchmark work offers insights for other eukaryote genomes such as mildew and microalgae, and even complicated human genomes, in addition to helping to build the high-quality genomes of Lepidopteran pests.

## 4. Materials and Methods

### 4.1. Insect Material

*B. mori* strains (P50T, D9L, and D9 × N4) were sourced from the Center for Frontier Interdisciplinary Biology, Southwestern University, China. Silkworms were reared on mulberry leaves under 12-h light and 12-h dark photoperiod at 28 °C from the 1st to 4th instars, and 25 °C after the 4th ecdysis.

### 4.2. Genome Sequencing and Creation of Subsets

Genomic DNA of P50T, D9L, and D9 × N4 was extracted, detected, and sequenced for generating PacBio HiFi, PacBio CLR and ONT reads at Frasergen (Wuhan, China), separately. Among them, D9L and D9 × N4 have not been previously sequenced.

Appendix A provides a summary of the statistics for each sequenced dataset. In order to investigate the dependence of assemblers on different sequencing depths and its influence on the quality of assembly, we used Seqtk (v1.2) and randomly selected eight subsets with divergence sequence depths (10×, 20×, 40×, 60×, 80×, 100×, 120×, 160×) in ONT data, seven subsets (the depths were 10×, 20×, 40×, 60×, 80×, 100×, 110×) of CLR data and six subsets (the depths were 10×, 20×, 30×, 40×, 50×, 60×) in HiFi data. Each subset shared similar read length distribution and coincident read length density (Figure 1).

### 4.3. De Novo Genome Assembly Workflow

De novo genome assembly and polishing workflow are displayed in Figure 2. For ONT and CLR subsets, we used four long-read assembly tools with default parameters: Canu (v1.9) [32], wtdbg2 (v2.5) [33], NECAT (v20200803) [34] /MECAT2 (v20200228) [35], and NextDenovo (v2.5.0) (https://github.com/Nextomics/NextDenovo, accessed on 10 June 2022). For HiFi subsets, we performed two assemblers, HiCanu (v2.2) [36] and hifiasm (v0.16.1) [37] using the default parameters. The long-reads were mapped to the graft assemblies with Minimap2 (v2.17) [38] and then polished using Racon (v1.5.0) [39].

### 4.4. Hi-C Scaffolding and Gap Filling

The Hi-C raw data of silkworm (P50T) was used to scaffold the genome assembly to the chromosomal level. Low-quality Hi-C raw reads were filtered out using Trimmomatic (v0.39) [40] (LEADING:3 TRAILING:3 SLIDINGWINDOW:4:15 MINLEN:50 CROP:50). The clean paired-end reads were mapped to the draft assembly by bwa (0.7.17) [41], and then analyzed by juicer (v1.6) [42]. Following this, 3D-DNA (v 180419) [43] and juicerbox (v1.11.08) [44] was applied to produce the chromosome-level assembly for silkworm. TGS-GapCloser (v 1.1.1) [45] was used to close gaps in the genome assemblies by long-reads.

### 4.5. Genome Assembly Evaluation

QUAST (v5.0.2) and Inspector were used to assess the assembly quality generated by different assembly tools. BUSCO(v4) was used to assess the completeness of the genome assembly with the insect (odb10) protein set. We selected the number and N50 of contigs, complete genes number from BUSCO, Small-scale errors per Mb, the number of Structural errors, and Quality Value (QV) score from Inspector to visualize in the main text. The QV was calculated on the base of the identified structural and small-scale errors in the assemblies [18].

Furthermore, we designed a quality assessment standard for chromosome-level genome assembly based on EagleC, a deep-learning framework for detecting a full range of assembly errors from Hi-C contact maps that were used to identify both small-scale and large-scale assembly errors, accurately. It reports the percentage (Equation (1)), type, and locus of specific assembly errors, and provides solutions on how to fix these assembly errors.
(1)$$\mathrm{Percentage}\;\mathrm{of}\;\mathrm{assembly}\;\mathrm{errors}\;\left(\mathrm{PAR}\right)=\frac{\;\mathrm{Total}\;\mathrm{length}\;\mathrm{of}\;\mathrm{the}\;\mathrm{assembly}\;\mathrm{errors}}{\;\mathrm{Total}\;\mathrm{length}\;\mathrm{of}\;\mathrm{the}\;\mathrm{assembly}\;}$$

### 4.6. Chromosomal Synteny Analysis

Genome comparisons were completed using NUCmer (v4) [46] with default parameters. NUCmer’s alignment file was filtered using delta-filter(-i 85 -l 8000 -o 85 -1). Mummerplot was used to create a dot plot. TBtools [47] was used to create a circos plot.

## Figures and Tables

**Figure 1 ijms-24-00649-f001:**
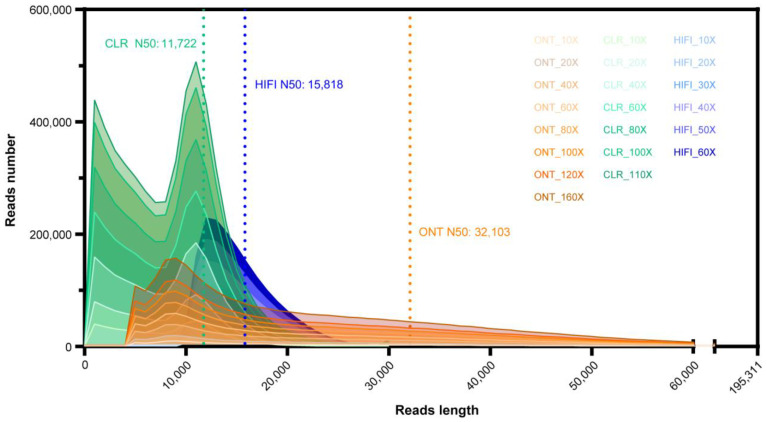
De novo assembly subsets with different data depths.

**Figure 2 ijms-24-00649-f002:**
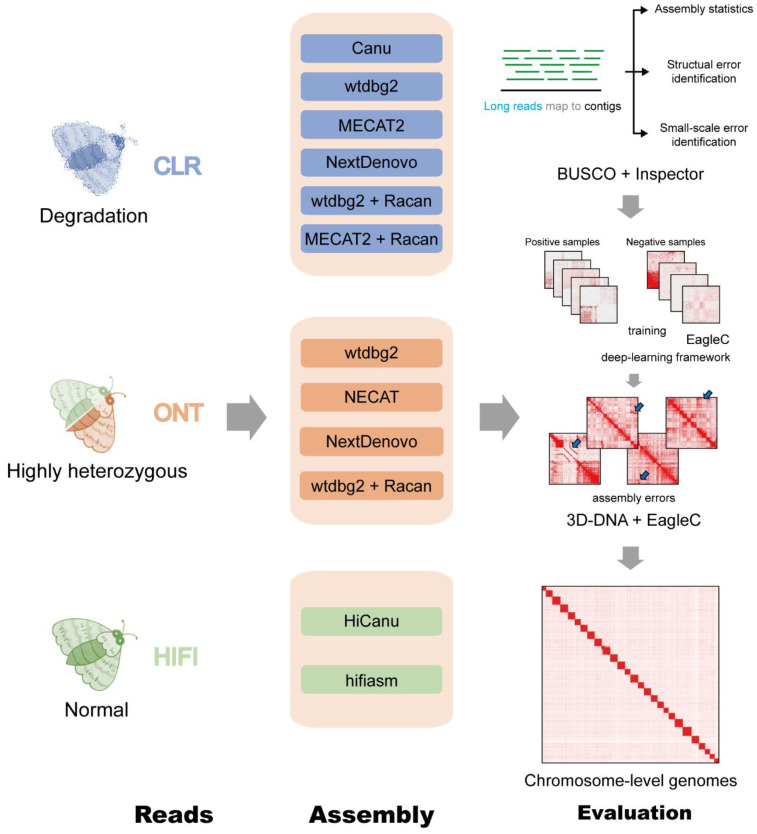
Summary of de novo assembly workflow and evaluation.

**Figure 3 ijms-24-00649-f003:**
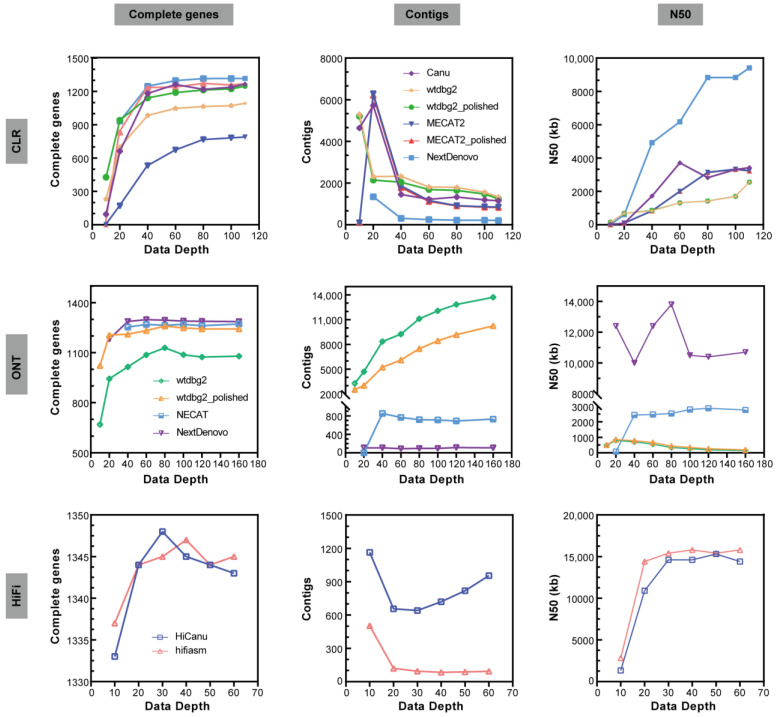
Main metrics of assemblies on CLR, ONT, HIFI subsets with different data depths. Complete gene number from BUSCO (Complete genes), contig number (Contigs), and N50 of contigs (N50).

**Figure 4 ijms-24-00649-f004:**
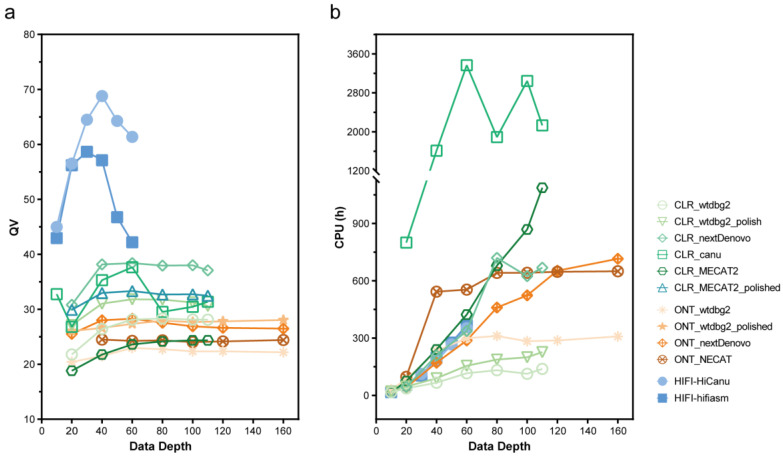
Quality Value (QV) score and computational time of assemblies on CLR, ONT, HIFI subsets with different data depths. (**a**) QV and (**b**) Computational time of different assemblers.

**Figure 5 ijms-24-00649-f005:**
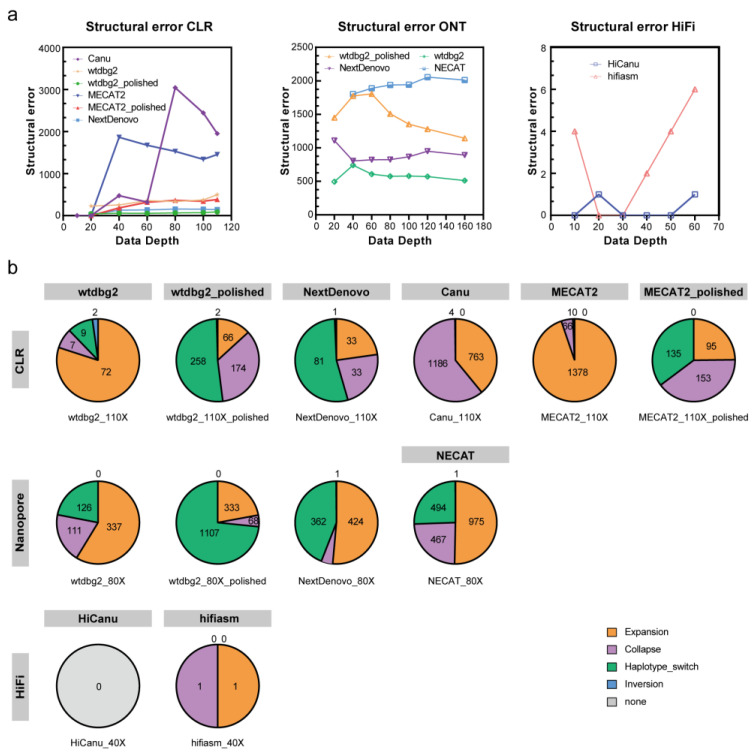
Structural error of assemblies on CLR, ONT, HIFI subsets. (**a**) Structural errors of assemblies with different data depths. (**b**) Pie graphs displaying the percentage of four types of structural errors discovered by the Inspector. The number of assembly errors is shown in each sector.

**Figure 6 ijms-24-00649-f006:**
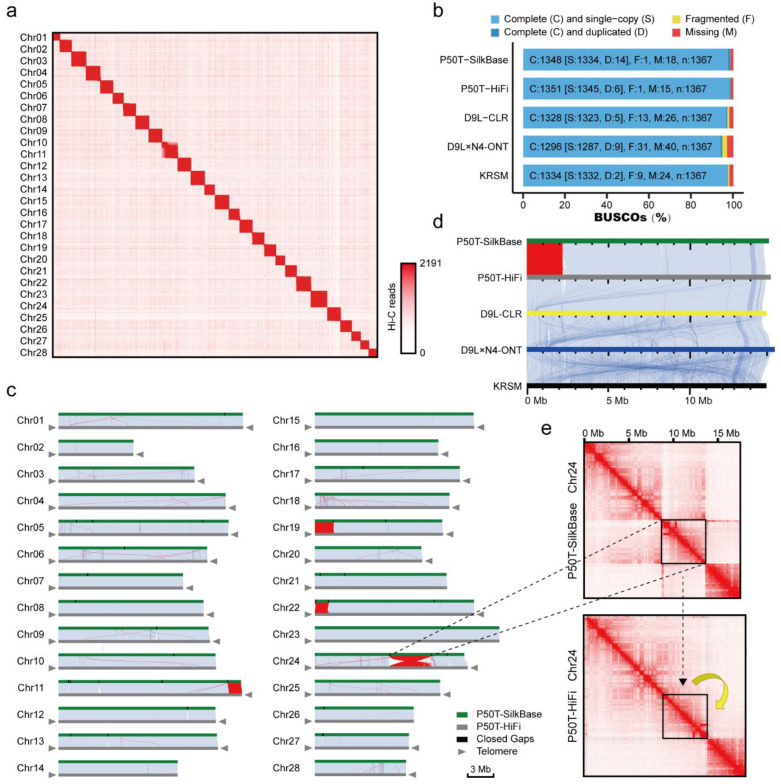
Summary of different silkworm strains chromosome-level genome assembly. (**a**) Hi-C genome-wide interaction map of the silkworm (P50T-HiFi) assembly. (**b**) BUSCO analysis of chromosome-level genome assemblies using the insect odb10 (1367 genes). (**c**) Collinearity between the silkworm P50T-SilkBase and P50T-HiFi genomes. The synteny blocks are shown by light blue lines. The inversions are indicated by red lines. The telomere sequence repeats are marked by gray triangles. All the P50T-SilkBase gap regions closed in P50T-HiFi are shown as black blocks. The gray triangles indicate the presence of telomere sequence repeats. (**d**) Collinearity of the P50T-SilkBase, P50T-HiFi, D9L-CLR, D9L × N4-ONT and Korean silkworm (KRSM) Chromosome 19. Synteny blocks are shown by light blue lines. The inversions are indicated by red lines. (**e**) Hi-C interaction map of silkworm chromosome 24. The black boxed area indicates the error inversion in P50T-SilkBase.

**Figure 7 ijms-24-00649-f007:**
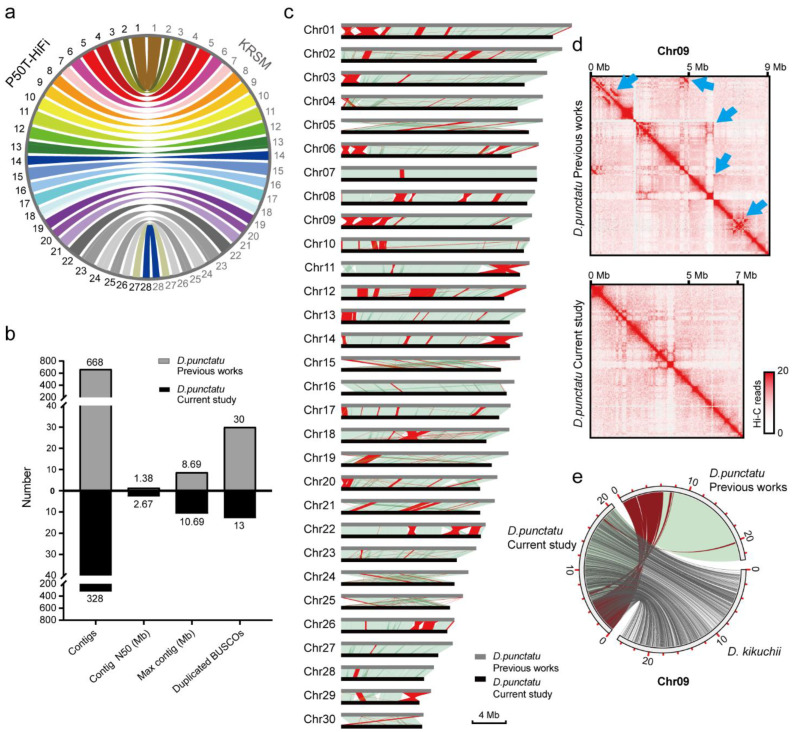
Summary of KRSM and *Dendrolimus punctatus* (*D. punctatus*) assemblies. (**a**) The synteny between silkworm P50T-HiFi and KRSM genomes. (**b**) Comparative analysis of the metric values of *D. punctatus* genome assemblies between the previous studies and this study. Contig numbers (Contigs), N50 of contigs (Contig N50), the maximum length of contigs (Max contig) and duplicated complete BUSCO gene number (Duplicated BUSCOs). (**c**) Collinearity of *D. punctatus* genome assemblies in previous research and this study. The synteny blocks are shown by light green lines. The inversions are indicated by red lines. (**d**) Hi-C interaction map of *D. punctatus* chromosome 9. The assembly errors in previous research were marked by blue arrows. (**e**) Collinearity circle plots. The synteny blocks are shown by light blue and gray lines. The inversions are indicated by red lines. It showed that the genome of *D. punctatus* (in this study) shared good collinearity with *Dendrolimus kikuchii* (*D. kikuchii*), and confirmed the assembly errors in previous studies.

**Table 1 ijms-24-00649-t001:** Sequence datasets analyzed in this study.

Acronym	Description	Reference
CLR	PacBio Continuous Long Reads; 48 Gb (110× coverage, E-size = 11,909 bp)	This study
ONT	Oxford Nanopore Technologies Reads; 70 Gb (160× coverage, E-size = 33,543 bp)	This study
HIFI	PacBio High Fidelity reads; 27 Gb (60× coverage, E-size = 16,484 bp)	This study
Hi-C	High-throughput Chromosome conformation capture sequencing; used for scaffolding	Lu et al. (2020) [20]

**Table 2 ijms-24-00649-t002:** The mean of metric values of different assemblers on CLR, ONT, and HIFI datasets.

	Contigs ^a^	Contig N50 (kb)	Structural Error	Small-Scale Error (/Mb)	QV ^b^	Complete Genes ^c^
CLR-Canu	2387.9	2161	1177.3	515.3	32	989.1
CLR-wtdbg2	2347.3	1236	65.2	2386.8	26.8	883.9
CLR-wtdbg2_polished	2206.6	1246	342	765.7	30.6	1054.1
CLR-MECAT2	1992	2111	1310	5594.9	22.8	618
CLR-MECAT2_polished	1948.5	2106	265.5	490.9	32.4	1182
CLR-NextDenovo	424.2	6463	125	207.4	36.7	1237.5
ONT-wtdbg2	9410.5	442	582.4	4488	22.1	1010.1
ONT-wtdbg2_polished	6526	516	1473.3	788.6	27.4	1207.6
ONT-NECAT	742.8	2656	1939.2	1691	24.3	1264.8
ONT-NextDenovo	103.1	11,454	895.3	1093.5	27.1	1275
HIFI-HiCanu	825.5	11,854	0.3	5.6	66.1	1342.8
HIFI-hifiasm	163.7	13,247	2.7	9.4	50.6	1343.7

**^a^** contig numbers, **^b^** Quality Value (QV) score from Inspector, **^c^** Complete BUSCO genes numbers.

**Table 3 ijms-24-00649-t003:** Statistics of different silkworm chromosome-level genome assemblies.

	N9L-CLR ^a^	D9L × N4-ONT ^b^	P50T-HiFi ^c^	KRSM ^d^
Size (Mb)	446.7	454.6	456.6	446.2
Scaffold N50 (Mb)	16.92	17.48	16.92	16.89
Contigs	51	29	29	33
Contig N50 (Mb)	14.38	17.48	16.92	16.89
Max contig ^e^ (Mb)	21.50	21.92	21.53	21.51
Structural error	193	852	2	34
Small-scale error (/Mb)	88.47	980.86	1.79	55.13
QV	36.84	27.37	56.84	41.75
BUSCOs ^f^	97.2%	94.8%	98.8%	97.5%
PAR ^g^	0.11%	0.03%	0.03%	0.05%

^a–d^ chromosome-level genome assemblies of silkworm N9L, D9L × N4, P50T and KRSM, ^e^ Maximum length of contigs, ^f^ Percentage of Complete BUSCO genes, ^g^ Percentage of assembly errors identified by EagleC.

## Data Availability

All raw sequencing data are available from the NCBI database with BioProject accession PRJNA898773. The genome assemblies of silkworm strain P50T, D9L and D9L × N4 are available from the NCBI database with BioProject accession PRJNA912164, PRJNA912165 and PRJNA912157.

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
