# Peer review of "Comparison of Long-Read Methods for Sequencing and Assembly of Lepidopteran Pest Genomes"

_ijms, 2022, doi:10.3390/ijms24010649_

Round 1
Reviewer 1 Report
It is an excellent study and indicates how to obtain and evaluate high-quality genome assemblies, and is a resource for biological control, comparative genomics, and evolutionary studies of lepidopteran pests and related species.
Author Response
Response to Reviewer 1 Comments
Dear reviewer #1,
Thank you for your recognition of our study. It is great honor to been reviewed and commented by you. We have uploaded the revised manuscript. The revised portions are marked in yellow in the version of the manuscript with the changes highlighted.
Comment: It is an excellent study and indicates how to obtain and evaluate high-quality genome assemblies, and is a resource for biological control, comparative genomics, and evolutionary studies of lepidopteran pests and related species.
Response : Special thanks to you for your good comments.
Reviewer 2 Report
The authors present the " first nearly complete 18 telomere-to-telomere reference genome of silkworm (P50T) produced by PacBio HiFi sequencing". This is good news, but there were already many good Bombyx mori assemblies available. Additional data and genomic improvements are always welcome IMO, so it should be a valuable addition. The data look good, and should be available in public repositories. I saw that the raw reads were deposited in a public repository, but the assemblies are not. The statement "The corresponding author can provide all the created assemblies for this work upon request." is quite shameful for a scientist in the modern age. These should be uploaded to a site such as Dryad or Zenodo.
That the main product from the manuscript is only "available upon request" reflects very badly on the manuscript. It need to be uploaded and publicly available or the paper falls flat on its face.
The scientific name (Bombyx mori) should be in the abstract.
There are several statements that cannot be understood, or may be misleading, without additional context. The authors state "For PacBio CLR and ONT sequencing, NextDenovo 23 is superior, and for PacBio HiFi, hifiasm is better. Meanwhile, we recommend using 3D-DNA with 24 EagleC evaluation to construct high-quality genomes." So they first state that two softwares are objectively better, and provide their recommendation on a third. These statements, and many others throughout the manuscript, need to be coherently qualified.
Similar to the context issues, the statements are often unlinked and flow could be greatly improved. For example, the authors state that "The greatest difference is reflected in the contig numbers, which are much smaller in hifi- 228 asm assemblies than in HiCanu assemblies. Then, we polished the HiFi assemblies using 229 the HiFi long-read sequences." The "Then" does not follow any action in the previous sentence. The whole manuscript reads like this, so attention to flow and correct English usage needs to be established.
Reviewer 3 Report
In this study, authors used silkworm Bombyx mori as a model to investigate genome sequencing and assembly through high-coverage datasets by de novo assemblies. Authors assessed the performance of diverse third-generation sequencing approaches in B. mori, focusing on how to efficiently and accurately construct and evaluate chromosome-level genome assemblies in B. mori and other Lepidopteran pests. This work will provide valuable guidance for future Lepidopteran pest genome projects as well as improve previous genome assemblies without generating new sequencing data.
Line 97 and 112, D9L×N4 or D9×N4?
Line 100, third-generation sequencing.
Line 298 and 308, abbreviations should be defined at first mention and used consistently thereafter.
Line 330, Limitations of the work must be mentioned in the discussion section.
Line 392, the font size is different from full text.
Line 531, the reference format is different with others.
Round 2
Reviewer 2 Report
The authors have responded to my comments appropriately and I am satisfied with the work.